# Redefining Accuracy: Underwater Depth Estimation for Irregular Illumination Scenes

**DOI:** 10.3390/s24134353

**Published:** 2024-07-04

**Authors:** Tong Liu, Sainan Zhang, Zhibin Yu

**Affiliations:** 1Key Laboratory of Ocean Observation and Information of Hainan Province, Sanya Oceanographic Institution, Ocean University of China, Sanya 572024, China; liutong_1106@stu.ouc.edu.cn (T.L.); zhangsainan@stu.ouc.edu.cn (S.Z.); 2Faculty of Information Science and Engineering, Ocean University of China, Qingdao 266100, China

**Keywords:** underwater monocular depth estimation, underwater image enhancement, auxiliary underwater depth information, self-supervised network

## Abstract

Acquiring underwater depth maps is essential as they provide indispensable three-dimensional spatial information for visualizing the underwater environment. These depth maps serve various purposes, including underwater navigation, environmental monitoring, and resource exploration. While most of the current depth estimation methods can work well in ideal underwater environments with homogeneous illumination, few consider the risk caused by irregular illumination, which is common in practical underwater environments. On the one hand, underwater environments with low-light conditions can reduce image contrast. The reduction brings challenges to depth estimation models in accurately differentiating among objects. On the other hand, overexposure caused by reflection or artificial illumination can degrade the textures of underwater objects, which is crucial to geometric constraints between frames. To address the above issues, we propose an underwater self-supervised monocular depth estimation network integrating image enhancement and auxiliary depth information. We use the Monte Carlo image enhancement module (MC-IEM) to tackle the inherent uncertainty in low-light underwater images through probabilistic estimation. When pixel values are enhanced, object recognition becomes more accessible, allowing for a more precise acquisition of distance information and thus resulting in more accurate depth estimation. Next, we extract additional geometric features through transfer learning, infusing prior knowledge from a supervised large-scale model into a self-supervised depth estimation network to refine loss functions and a depth network to address the overexposure issue. We conduct experiments with two public datasets, which exhibited superior performance compared to existing approaches in underwater depth estimation.

## 1. Introduction

An underwater depth map, serving as a visual representation or data channel, provides essential information about the spatial distances of objects within an underwater scene from a specific viewpoint [1]. Underwater depth maps involve various applications, including underwater autonomous navigation [2], marine archaeology [3], simultaneous localization and mapping (SLAM) [4], and so on. While most current depth estimation methods focus on estimating depth by analyzing the optical information of scenes, they do not consider irregular illumination in underwater scenes, which is very common in underwater environments due to low-light conditions and overexposure. Improving the accuracy of underwater depth map estimation in irregular illumination scenes remains a significant research direction.

Exploring depth information involves the utilization of various technologies, such as LiDAR [5] and Kinect [6]. LiDAR systems use lasers to actively illuminate their surroundings and measure distances by calculating the time it takes for laser reflections to return [7]. Kinect, a discontinued line of motion sensing devices, contains a four-microphone array, a color camera, and a depth sensor that acquires depth information by structured light or time-of-flight (ToF) calculations [8]. Nonetheless, most of these active depth sensing methods can be easily influenced by underwater characteristics such as scattering and absorption [9]. Although sonar-based devices can be used to obtain underwater 3D information [10], sonar-based technologies can hardly obtain an accurate depth map due to the limitation of bandwidth [11]. Prior-based methods, such as dark channel prior (DCP) [12], are another solution for estimating underwater depth maps. Muniraj et al. obtain a transmission depth map based on the difference between the maximum and minimum intensities prior [13]. Physical-model-based depth estimation methods analyze regions of low pixel values in images to obtain important information about illumination and scene structure. However, these methods generate depth maps by establishing the relationship between transmission maps and depth maps. Unknown scattering parameters and various factors complicate depth estimation [14]; therefore, depth estimation based on physical models becomes challenging in low-light conditions, as shown in Figure 1.

In recent years, the advancement of deep learning has propelled the application of computer vision, including advancements in tasks such as object detection [16], semantic segmentation [17], depth estimation [18], and beyond. In the domain of depth estimation, self-supervised learning has garnered attention due to its ability to train models without the need for depth ground truth labels [19], as well as its capacity to explore spatial relationships within the training data [20]. It is within this situation that MonoDepth2 [18] emerges. Its role as a widely used self-supervised depth estimate pipeline demonstrates a certain level of generalizability [9]. Based on the architecture, MonoViT [21] and Lite-Mono [22] employ vision Transformers to enhance the receptive field of convolutional neural networks (CNNs). Indeed, these attention-based self-supervised monocular depth estimation methods primarily focus on improving depth estimation accuracy by learning more features from the dataset. Through data augmentation, Saunders et al. [23] improve depth estimation accuracy in adverse scenarios. However, these deep-learning-based self-supervised methods can hardly account for depth estimation in overexposed underwater scenes. Since these methods only consider optical constraints without geometric constraints, a distortion caused by overexposure can break the optical constraints between frames. We present examples of two typical self-supervised monocular depth estimation methods with optical constraints applied in overexposed underwater scenes, as shown in Figure 2. Both MonoDepth2 [18] and MonoViT [21] exhibit poor performance in depth estimation, depicted in Figure 2.

We propose an underwater self-supervised monocular depth estimation network that integrates image enhancement and auxiliary depth information to address the depth estimation problem in underwater irregular illumination scenes. The main framework of our network is shown in Figure 3. Inspired by PUIE-Net [24], we employ a probability-based image enhancement method to provide better stability and effectively remove the interference caused by low-light factors in underwater scenes. In addition, the underwater overexposure phenomenon can break the optical constraints that widely exist in many underwater depth estimation frameworks. To address this issue, we employ a monocular depth estimation model [25] to provide additional geometric constraints for better depth estimation. In short, the main contributions of this paper can be summarized as follows:

We introduce the Monte Carlo image enhancement module (MC-IEM) to remove the interference caused by underwater low-light conditions and enhance depth estimation accuracy.We employ an auxiliary depth module (ADM) to provide extra geometric constraints to address the issue of distorted surface textures caused by overexposure between frames in underwater environments.We conduct extensive comparative experiments on two public underwater datasets. The experimental results demonstrate that our method surpasses other methods in the qualitative and quantitative sections.

## 2. Related Work

### 2.1. Physics-Based Methods

Some methods based on physical models restore images by estimating transmission parameters within the medium, and, during this process, depth maps are generated as secondary products [26]. Dark channel prior (DCP) [12] adopts the light characteristics for image enhancement and restoration. The depth maps, called transmission maps, are by-products of image enhancement and restoration methods. Peng et al. [27] employed depth maps as an intermediate step for image enhancement by analyzing image blurriness. Drews et al. [28] proposed a method that simultaneously restores underwater medium transmission, scene depth, and image visual quality using a combination of a physical model and scene statistical priors. As an extension of DCP, the authors of underwater dark channel prior (UDCP) [29] indicated that the blue and green channels should be considered independently to obtain more reliable transmission maps. The method proposed by Peng et al. [30] utilizes an image formation model (IFM) specific to submerged environments, addressing image blurriness and light absorption to estimate depth maps. Song and his team [31] introduced a swift and efficient model for estimating scene depth in underwater images, utilizing underwater light attenuation prior (ULAP). Berman et al. [32] considered various spectral profiles associated with different water types and achieved this by estimating only two additional global parameters. Bekerman et al. [33] generated depth maps by reconstructing a comprehensive physical model of the scene, incorporating estimated attenuation ratios and veiling light. Muniraj et al. [13] proposed another depth estimation method, the difference in channel intensity prior (DCIP), where depth maps are estimated based on the difference between maximum and minimum intensity priors. These physical-model-based depth estimation methods analyze the distribution of image pixels to estimate depth. However, as shown in Figure 1, these depth estimation methods may encounter difficulties in low-light underwater scenes. To address this issue, we employ the MC-IEM to eliminate the interference caused by low-light conditions (Figure 1e).

### 2.2. Deep-Learning-Based Methods

The development of deep learning has dramatically propelled the advancement of depth estimation. Deep-learning-based monocular depth estimation involves training a deep neural network to infer depth maps from color images [34]. Unlike the in-air depth estimation tasks, collecting RGBD-paired data is tough in underwater scenes. Due to the lack of paired underwater RGBD data, UW-Net [14] employs CycleGAN for style transfer to generate underwater-style RGBD images, thereby facilitating depth estimation in underwater scenarios. Ye et al. [35] introduced a novel architecture for joint underwater depth estimation and color correction, emphasizing an unsupervised adaptation network at style and feature levels. UW-Depth [36] obtains depth maps and synthesizes RGBD pairs simultaneously. Finally, a fine-tuning strategy and depth loss made it a more effective underwater depth estimation model. Taking advantage of unpaired image synthesis techniques and in-air paired RGBD images, the above methods tackled the problem caused by the lack of underwater paired RGBD data. However, the synthesized RGBD pairs still lack underwater objects, resulting in a poor performance of depth estimation models in natural underwater environments [1].

Utilizing self-supervised methods allows models to infer scene depth information from a single camera without manually annotated data, thus improving the applicability of depth estimation [18]. Methods for self-supervised depth estimation [18,37,38,39,40,41,42,43] based on monocular video sequences predominantly leverage photometric consistency loss to optimize the models. Godard et al. [18] employed minimum reprojection loss across frames and utilized multi-scale resolutions to estimate depth maps. SC-Depth V3 [44] improved monocular depth estimation in dynamic scenes using dynamic region refinement and local structure refinement modules, producing accurate depth maps in highly dynamic monocular video scenarios. MonoViT [21] utilized a self-attention mechanism to enhance the depth estimation capability of the model. Using a self-attention mechanism, Lite-Mono [22] adopts a lightweight backbone. Robust-Depth [23] achieves stable depth estimation through data augmentation. Chen et al. [45] addressed the widespread issue of edge fattening in self-supervised monocular depth estimation models by redesigning the patch-based triplet loss. MonoFormer [46] is a self-supervised depth estimation network consisting of a CNN–Transformer hybrid network and a multi-level adaptive feature fusion module. Since the self-supervised learning architectures have achieved tremendous success in many in-air depth estimation tasks [18,37,38,39,40,41,42,43], Yang et al. [9] proposed underwater self-supervised monocular depth estimation, focusing on scattering and absorption in underwater scenarios. Self-supervised depth estimation models use monocular video sequences to generate reliable depth maps and execute underwater tasks. However, these methods overlook the issue of the impact caused by overexposure. The distortion of surface textures caused by overexposure disrupts the optical constraints of depth estimation. We employ transfer learning with an auxiliary depth module (ADM) to extract additional geometric features to address this challenge.

## 3. Methods

### 3.1. Overall Framework

The general self-supervised monocular depth estimation pipeline mainly consists of a depth net and a pose net, constrained by multi-view consistency [44]. As shown in Figure 3, our network architecture includes a Monte Carlo image enhancement module (MC-IEM), an auxiliary depth module (ADM), a depth net, and a pose net. In our approach, the MC-IEM leverages probabilistic estimation to learn the inherent uncertainty in underwater light variations, enhancing the natural relationship among objects in the scene and mitigating the impact of low-light conditions. Specifically, we utilize consecutive frames from video sequences, denoted as $I_{t-1}$ and $I_{t}$, as inputs to MC-IEM. The outputs of this module are represented as $I_{t-1}^{\prime }$ and $I_{t}^{\prime }$. In addition, recent depth estimation methods encounter challenges due to overexposure, which distorts surface textures on submerged objects and affects the optical constraints of depth estimation models. To address this issue, we apply transfer learning to extract additional geometric features. This method incorporates prior knowledge from a supervised large-scale model [25] into our self-supervised monocular depth estimation network to refine losses. In our work, the ADM plays a significant role in acquiring additional depth information. The symbol *D* represents the output of the module. The results of depth estimation on $I^{\prime }$ are denoted as $D^{\prime }$, with $D_{t-1}^{\prime }$ and $D_{t}^{\prime }$ being the specific outputs. The pose net focuses on estimating the pose transformation between adjacent frames. The network constrains the pose net through photometric consistency loss. The accuracy is improved by constraining the depth net and pose net using edge-aware smoothness loss, normal matching loss, and depth gradient loss.

**Monte Carlo Image Enhancement Module (MC-IEM).** To address the challenges of light attenuation in underwater environments with low-light conditions, we utilize a Monte Carlo probabilistic estimation method. This method effectively manages the inherent uncertainty in such images, resulting in more precise visual representations. Additionally, we introduce a multi-scale strategy to ensure that no information is lost from the original images during the enhancement process. The strategy processes underwater images as inputs to a U-Net image enhancement network, which utilizes pre-trained weights from PUIE-Net [24] to learn the Gaussian distribution between enhanced and non-enhanced images. We then employ a consensus process to predict deterministic results based on samples from the distribution. This probability estimation and consensus process resolve bias in reference maps during image enhancement, thereby addressing inaccuracies caused by difficulties in object recognition under low-light conditions in depth estimation. We illustrate the image enhancement process of MC-IEM in Figure 4.

We incorporate a multi-scale strategy to process underwater images at different resolutions, allowing the network to capture fine details and more contextual information. MC-IEM utilizes prior knowledge of underwater images to estimate a Gaussian feature distribution ranging from blurry to clear. By randomly sampling *m* images from Gaussian distribution, we generate a series of potential images with varying levels of clarity and contrast, denoted as $I_{ti}^{\prime }$, where i=1…m. These sampled images are then aggregated to compute the final enhanced images $I_{t}^{\prime }$ through a consensus process that reduces random errors by integrating information from multiple images. This step results in visually higher contrast and clearer final enhanced images. The following formula explains MC-IEM [24]:(1)$$p\left(I_{t}^{\prime }|I_{t}\right)\approx \frac{1}{m}\sum_{i=1}^{m}p\left(I_{t}^{\prime }|z,I_{t}^{S}\right),z∼p\left(z|I_{t}^{S}\right),$$

The input to the network consists of three scaled images, each with a scale of 0.5, 1, or 1.5. These images are designated as $I_{t}^{S}$. To express uncertainty, we make use of an implicit variable *z*. *z* represents camera/algorithm parameters or human-subjective preferences in capturing the ground truth. Let $I_{t}$ denote the corrupted observation. $I_{t}^{\prime }$ represents the enhanced image. Here, $p(z|I_{t}^{S})$ signifies the uncertainty distribution and *m* denotes the number of samples.

**Auxiliary Depth Module (ADM).** To mitigate the impact of overexposure on most current self-supervised monocular depth estimation methods, which rely on optical information to constrain consecutive frames, we introduce an auxiliary depth module. This module works by extracting additional geometric features from a supervised large-scale model [25]. These features are then integrated into the self-supervised monocular depth estimation pipeline, guiding the model to generate more accurate depth maps through depth gradient loss and normal matching loss. Additionally, the auxiliary depth module regularizes the depth estimation process, reducing sensitivity to overexposure variations. This comprehensive approach significantly enhances the robustness and accuracy of self-supervised monocular depth estimation, particularly in overexposure conditions.

### 3.2. Loss Functions

**Depth Gradient Loss.** We propose a depth gradient loss based on geometric consistency constraints to address photometric inconsistencies between consecutive frames caused by overexposure. We constrain the network by enforcing geometric consistency and normalizing the outputs from the depth network and the auxiliary depth module. We define our depth gradient loss as $L_{D}$:(2)$$L_{D}=\frac{1}{N}\sum_{i=1}^{N}\|G_{ti}-G_{ti}^{\prime }{\|}_{1},$$
where *N* represents the count of gradient pixels. The depth map $D_{t}^{\prime }$, generated by the depth net, is transformed into a gradient map $G_{t}^{\prime }$, while the depth map $D_{t}$, produced by the ADM, is transformed into a gradient map $G_{t}$.

**Normal Matching Loss.** In response to the challenge posed by overexposure scenes, which can disrupt essential geometric constraints for self-supervised monocular depth estimation methods, our approach leverages both the auxiliary depth maps generated by the ADM and the depth net. We enhance the geometric constraints by computing normal maps from these depth maps. This step is essential for ensuring that the predicted depth maps closely aligns with the geometric structures. Our proposed normal matching loss, $L_{N}$, is inspired by prior work of Libo Sun et al. [44] and has been designed to address the challenges posed by overexposed underwater scenes:(3)$$L_{N}=\frac{1}{N}\sum_{i=1}^{N}\|n_{ti}-n_{ti}^{\prime }{\|}_{1},$$
where *N* represents the count of normal pixels. The depth map $D_{t}^{\prime }$, generated by the depth net, is transformed into a normal map $n_{t}^{\prime }$, while the depth map $D_{t}$, produced by the ADM, is transformed into a normal map $n_{t}$.

**Photometric Consistency Loss.** Following MonoDepth2 [18], we minimize color inconsistencies between adjacent underwater frames using intrinsic *K*, depth maps from depth net ($D^{\prime }$), and pose transformations from pose net (*P*). We describe $L_{P}$ as follows:(4)$$L_{P}=\frac{1}{\left|V\right|}\sum_{p\in V}M\left(p\right)(\lambda (\|I_{t}^{\prime }\left(p\right)-I_{t^{\prime }}^{\prime }{\left(p\right)\|}_{1})+\left(1-\lambda \right)\frac{1-SSIMtt^{\prime }\left(p\right)}{2}),$$
where λ is set as 0.15 and M(p) is a self-discovered mask [18] that reduces the influence of dynamic objects on underwater depth estimation by masking them within the scenes. $I_{t-1}^{\prime }$ and $I_{t}^{\prime }$ represent the previous frame and the current frame, respectively. $I_{t^{\prime }}^{\prime }$ is obtained by combining intrinsic *K*, depth map $D_{t}^{\prime }$, and pose $P_{tt-1}$. The expression for $I_{t^{\prime }}^{\prime }$ is provided in Equation (Equation 5), following MonoDepth2 [18]: (5)$$I_{t^{\prime }}^{\prime }=I_{t-1}^{\prime }\langle \mathrm{proj}\left(D_{t}^{\prime },P_{tt-1},K\right)\rangle ,$$

**Edge-aware Smoothness Loss.** Following MonoDepth2 [18], we employ the edge-aware smoothness loss to smooth the edge regions of the estimated depth maps. The expression for $L_{S}$ can be written as follows:(6)$$L_{S}=|\partial_{x}d_{t}^{*}|e^{-|\partial_{x}I_{t}|}+\left|\partial_{y}d_{t}^{*}\right|e^{-|\partial_{y}I_{t}|},$$
where $d_{t}^{*}$ is defined as the ratio $d_{t}^{*}=D_{t}^{\prime }/\bar{D_{t}^{\prime }}$. The $\bar{D_{t}^{\prime }}$ represents the mean of the depth values generated by the depth net.

**Final Loss.** Combining all the loss functions discussed above, our final loss is defined as follows:(7)$$L=\omega_{D}L_{D}+\omega_{N}L_{N}+\omega_{P}L_{P}+\omega_{S}L_{S},$$

In this setting, the weights for $\omega_{P}$ and $\omega_{S}$ follow the guidelines in MonoDepth2 [18]. Specifically, we assign the values $\omega_{P}=1.0$ and $\omega_{S}=0.1$. Meanwhile, our decision to set $\omega_{N}=0.2$ and $\omega_{D}=0.1$ is influenced by the insights presented in existing depth estimation models [38,39,44].

## 4. Results

### 4.1. Datasets and Experimental Details

In this section, we adopt two publicly available underwater datasets for our experiments: FLSea [47] and SQUID [32].

FLSea [47] comprises visual–inertial videos captured in two regions: the canyon and the Red Sea, totaling 12 scenes and containing 22,451 images, each with a resolution of 968×608. FLSea [47] includes underwater RGB images, depth maps, and camera intrinsic parameters. Camera intrinsic parameters and depth maps are obtained using Agisoft Metashape (version 2018) [48]. The scenes in FLSea [47] include low-light scenes, overexposed scenes, unevenly illuminated underwater scenes, and relatively ideal underwater scenes. According to our observation, the brightness between adjacent frames does not change significantly in scenes.

Specific scenarios for the FLSea [47] are shown in Figure 5.

In addition to testing with the FLSea [47], we employ SQUID [32] to validate the generalization of our self-supervised monocular depth estimation method. This dataset contains stereo images from four scenes: Katzaa, Michmoret, Nachsholim, and Satil. The dimensions of the images in this dataset are 5474×3653. Some examples of SQUID [32] are shown in Figure 6.

Regarding the specific setup details of our experiment, we employed a GeForce RTX 2080, sourced from NVIDIA in Santa Clara, CA, USA, with 8 GB of graphics memory. Our CPU includes dual Intel Xeon E5-4627 v4 processors. Each processor features 10 cores operating at a base frequency of 2.60 GHz, collectively supporting 20 threads. The length of the continuous sequence frame input during network training is 3. The input images for the network were resized to 256×320. Our experiment relied on the PyTorch library [49]. Following the setting of MonoDepth2 [18], we set the learning rate to $10^{-4}$ and experimented with 100 epochs. According to Figure 7, conducting 100 training epochs is optimal; however, further epochs lead to overfitting. When training the model for fewer than 100 epochs, our model has not yet achieved the best depth estimation performance and remains underfitted. Considering our available graphics memory, we set the batch size to 4. For each scene of FLSea [47], the final 200 frames of each scene were utilized for validation and testing purposes (the initial 50 frames for validation and the final 150 frames for testing). In contrast, the remaining frames were employed to train the self-supervised monocular depth estimation network. In conclusion, the training set included 20,051 images, the validation set included 600 images, and our testing set included 1800 images. Additionally, to validate the generalization of our method, we randomly selected 71 images from SQUID [32] for the quantitative experiments.

To assess the quality of our model, we adopted four evaluation metrics [18]. These metrics include mean absolute relative error (AbsRel), absolute relative logarithmic error (Log10), root mean squared logarithmic error (RMSElog), and accuracy measured with a threshold (δ<threshold).

### 4.2. Evaluation

#### 4.2.1. Qualitative Evaluation

In this section, we compare our model against some traditional methods (DCP [12] and UDCP [15]) and deep-learning-based methods (UW-Net [14], SC-Depth V3 [44], MonoViT [21], Lite-Mono [22], and Robust-Depth [23]).

We performed qualitative experiments on the test set of FLSea [47]. The final comparison results of state-of-the-art depth estimation methods are shown in Figure 8. We chose three images, including one low-light scene and two images with overexposure, for evaluation. To facilitate the observation of our results, we added a color bar depicting depths from 0 to 12 m, aligned with the depth range of the FLSea [47]. Figure 8b,c show the results of two physical-model-based depth estimation methods (DCP [12] and UDCP [15]). Both DCP and UDCP [12] perform poorly in low-light scenes. The light and dark pixels on the same rock show quite different depth estimation results (Figure 8b,c). In addition, we also chose some deep-learning-based methods for comparison, as shown in Figure 8d–h. UW-Net [14] can roughly provide depth estimation information for the second and third scenes but fails in the low-light case of the first image. SC-Depth V3 [44], MonoViT [21], Lite-Mono [22], and Robust-Depth [23] can provide roughly correct estimation results in the second scene. However, only MonoViT [21] achieves superior depth estimation results in both the first and third scenes due to its use of an attention mechanism. Other methods cannot accurately detect the rock in the center of the first and third scenes. We guess that the challenge comes from the overall low light in the first image and the overexposure in the left lower corner of the third image. The scattering surrounding the rock further increases the difficulty of estimation. Our method, which integrates the ADM and MC-IEM, performs best.

Furthermore, to validate the effectiveness and generalizability of our model, we also utilized SQUID [32] for additional qualitative comparisons. The specific results of the qualitative experiments are shown in Figure 9.

On SQUID [32], which has brighter underwater scenes compared to FLSea [47], we selected four images for qualitative demonstration. To observe the visualization results, we referred to SQUID’s [32] depth range and set the color bar from 0 to 15 m. Figure 9b,c generate poor depth map estimation due to the uncertain seawater scattering parameters, which impact the guidance provided by the dark channel information across the four images. The deep learning methods UW-Net [14] and SC-Depth V3 [44] can roughly estimate the distance of underwater scenes. Due to the strong scattering and limited feature extraction capabilities of these methods, UW-Net [14] and SC-Depth V3 [44] cannot estimate the depth of rocks accurately. MonoViT [21], Lite-Mono [22], and Robust-Depth [23] can provide more accurate depth information, with MonoViT [21] and Robust-Depth [23] particularly excelling at estimating the depth of rocks in the scenes. Due to the geometric consistency constraints between frames applied to overexposed conditions, our depth estimation method achieves the best depth estimation results even in bright scenes with scattering.

#### 4.2.2. Quantitative Evaluation

In the quantitative evaluation section, we adopt the metrics mentioned in Section 4.1, including error metrics AbsRel, Log10, and RMSElog [18], as well as the accuracy metric δ. We conducted quantitative experiments on FLSea [47]. The results are shown in Table 1. We highlight the best results in bold.

Consistent with the qualitative assessment, DCP [12] shows the greatest error in results, mainly because of uncertain physical parameters (such as scene radiance and scattering coefficient). UDCP [15] employs blue–green channel information for depth estimation, thereby outperforming DCP [12] in depth estimation. MonoViT [21] performs poorly on RMSElog, indicating inaccurate depth estimates in areas with large depth values, i.e., farther distances. Lite-Mono [22] outperforms our method by 0.088 in terms of RMSElog and by 0.003 in terms of $\delta <1.25^{3}$, suggesting more accurate depth estimation at farther distances. Compared with methods based on physical and deep learning models, our method achieves the best performance on most metrics, attributed to the stability provided by Monte Carlo image enhancement and the additional geometric features provided by ADM.

We conducted quantitative comparisons on SQUID [32], which obtains depth ground truth by SFM. However, the ground truth depth may not be densely populated across all image regions. Therefore, our evaluation focuses on depth values where ground truth information is valid. The results of our experiments are presented in Table 2.

As shown in Table 2, the depth estimation performance of all methods declines on cross-domain datasets. This deterioration can be attributed to the discrepancies between the training and test sets. Furthermore, Robust-Depth [23] outperforms other depth estimation methods in error and accuracy assessment. This superiority is attributed to its ability to achieve robust depth estimation results under challenging conditions by learning the mapping relationship between original and augmented scenes. The method significantly improves both AbsRel and RMSElog, as depicted in Figure 9. Our method achieves the best depth estimation results in the scenes due to our analysis of underwater illumination characteristics. We leverage the robust enhancement capability of MC-IEM and benefit from additional geometric constraints provided by ADM, contributing to improved depth estimation results.

#### 4.2.3. Ablation Study

We propose a self-supervised monocular depth estimation network based on a Monte Carlo image enhancement module and an auxiliary depth module. In order to verify the effectiveness of each module, we designed and executed ablation experiments, the details of which are shown in Figure 10.

We present four images to show the results of our ablation experiments on FLSea [47]: one low-light image, two overexposed images, and one image with an exposed foreground and low-contrast background. The color mapping is the same as that in Figure 8. As shown in the first and fourth images, our image enhancement module significantly improves depth estimation in low-light scenes, particularly in the foreground rock area of the first image and the left part of the canyon in the fourth image. Furthermore, our auxiliary depth module estimates depth in the overexposed sand of the second image and overexposed rock scenes of the third image through geometric constraints. Therefore, our method achieves a more accurate visual approximation to the ground truth. To rigorously demonstrate the effectiveness of our work, quantitative experiments were conducted, as shown in Table 3.

Incorporating the MC-IEM into the baseline results in a decrease in AbsRel and Log10 and an increase in RMSElog, indicating an overall improvement in depth estimation accuracy. However, the model still faces challenges in precisely estimating the depth of distant objects. As depicted in Figure 10, particularly evident in low-light scenarios, the Monte Carlo approach prioritizes the depth estimation of foreground objects. Furthermore, when we employed an auxiliary depth module to address the issue of overexposure present in the scene, there was an improvement in depth estimation. Absorbing prior knowledge from the large model effectively mitigates the issue of excessive exposure in underwater scenes, resulting in improved depth estimation, particularly in overexposed areas, and an overall improvement in depth estimation accuracy.

To further demonstrate the effectiveness of integrating auxiliary depth information into the self-supervised network, we compared our model with depth estimation methods based on supervised large models [25,50]. LeRes [51], DPT [50], and MIDAS [25] are supervised depth estimation methods. As shown in Table 4, MIDAS [25] generates comparatively superior depth maps compared to the other two estimation models. Therefore, we leverage the prior knowledge generated by MIDAS [25] to improve depth estimation in our underwater depth estimation network, which is achieved by incorporating inter-frame geometric constraints through our proposed depth gradient loss and integrated normal matching loss. However, as shown in Table 4, we do not employ MIDAS’s absolute depth for constraints. Absolute depth measures the distance from the camera to objects in the underwater scenes, covering distances ranging from 0 to 12 m in FLSea [47]. Scaling depth maps generated by MIDAS [25] to obtain absolute depth is inaccurate. Instead, we adopt normalized relative depth, which indicates the order of distances between objects in underwater environments [52], to provide additional geometric information about the scenes.

## 5. Discussion

### The Relationship between Image Enhancement and Depth Estimation

This section further explores the relationship between image enhancement and depth estimation. Many depth estimation methods based on physical models regard depth estimation as related to image enhancement [30]. In our study, we conducted many experiments to assess the impact of different image enhancement methods on depth estimation. As shown in Figure 11, we qualitatively compared the relationships between image enhancement methods and depth estimation on FLSea [47]. CLAHE [53] enhances detail in areas of low contrast by applying local histogram equalization. However, histogram equalization may result in an excessive enhancement of high-contrast or high-brightness areas of the image, which may result in the introduction of unnatural artifacts or color distortion. The self-supervised monocular depth estimation network performs depth estimation by learning the relationships and features between adjacent frames. When the right region of the image is over-enhanced by CLAHE [53], the consistency constraints between adjacent frames are reduced, affecting the accuracy of the depth estimation. In heavily degraded and texture-less underwater images, FUnIE-GAN [54] proves inadequate for enhancement. The resulting images frequently exhibit an exaggerated level of noise amplification, leading to image over-saturation. Despite generally accurate hue correction, color and texture recovery remain inadequate [54]. The original image in Figure 11 is a low-light underwater image. However, FUnIE-GAN does not accurately process the color and texture of this scene. Therefore, it is unrealistic to input such error-prone information directly into the self-supervised monocular depth estimation network and expect improved depth estimation results. Upon examining Figure 11c, it becomes evident that the FUnIE-GAN-enhanced image exhibits color issues.

Consequently, we have reason to suspect that this discrepancy may impact depth estimation accuracy. As mentioned above, we applied the same color mapping to FLSea [47]. Water-Net [29] is influenced by background scattering, resulting in inferior depth estimation for distant objects. As depicted in Figure 11d, depth estimation in the background region is also compromised, aligning with the observed image enhancement effect. The ground truth depth of this scene was obtained through COLMAP [48] in Figure 11f. Compared to other image enhancement methods, our approach utilizes an underwater enhancement technique based on probability and consensus processes with a multi-scale strategy. As a result, our image enhancement method enables the acquisition of accurate light information, thereby facilitating precise depth estimation.

We conducted quantitative experiments to compare the impact of different image enhancement methods on depth estimation. Table 5 provides the results of our quantitative comparison of different image enhancement methods’ impact on depth estimation.

Consistent with the qualitative comparison results in Figure 11, we find that CLAHE [53] is less effective for underwater images with color bias and poor depth estimation. By analyzing the quantitative metrics, we find that the logarithmic error of the depth estimation of FUnIE-GAN [54] is smaller, which indicates that FUnIE-GAN [55] is better at estimating long-distance scenes than Water-Net [29]. This is because Water-Net does not fully solve the problem of background scattering at long distances. Our method achieves the best depth estimation result due to the robustness of image enhancement in low-light scenes using Monte Carlo, coupled with the ability to capture more light information through a multi-scale strategy.

## 6. Conclusions

We propose an underwater self-supervised monocular depth estimation network that combines image enhancement and auxiliary depth information to address the issue of inaccurate depth estimation in irregular illumination scenes. To tackle the issue of reduced contrast in low-light environments, we employ a Monte Carlo image enhancement module (MC-IEM) based on probabilistic estimation. Secondly, overexposure can disrupt photometric consistency between frames. To address this challenge, our work integrates auxiliary depth information to constrain geometric consistency between frames by our proposed depth gradient loss and integrated normal matching loss. We conduct considerable experiments to demonstrate the effectiveness of our approach. The results demonstrate that our method performs best on two public underwater datasets. In addition, we plan to integrate our generated depth maps into underwater SLAM applications in future work. 

## Figures and Tables

**Figure 1 sensors-24-04353-f001:**
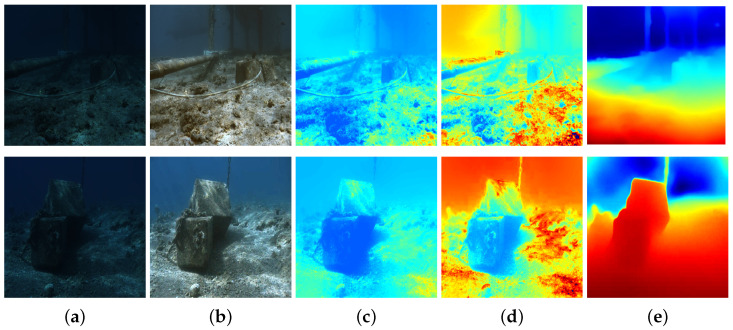
The comparison between the typical physical-model-based depth estimation methods and our work in low-light underwater environments. Red color indicates a close distance and the blue color represents a range distance. (**a**) Raw images. (**b**) The output of MC-IEM. (**c**) DCP [12]. (**d**) UDCP [15]. (**e**) Ours.

**Figure 2 sensors-24-04353-f002:**
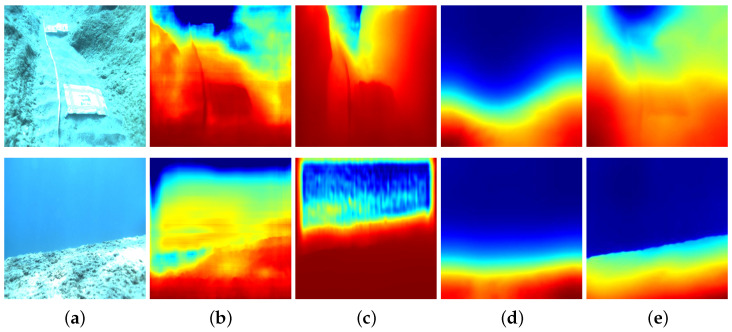
A comparison between depth estimation methods based on deep learning and our work in overexposed underwater environments. Red color indicates a close distance and the blue color represents a range distance. (**a**) Raw images. (**b**) MonoDepth2 [18]. (**c**) MonoViT [21]. (**d**) Ours without ADM. (**e**) Ours.

**Figure 3 sensors-24-04353-f003:**
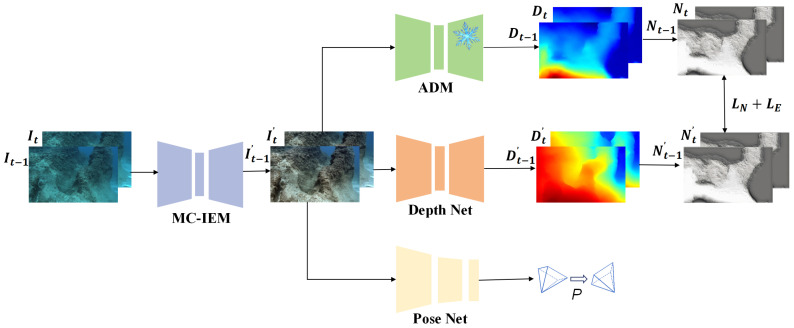
The network framework of our model consists mainly of a Monte Carlo image enhancement module (MC-IEM), an auxiliary depth module (ADM), a depth net, and a pose net.

**Figure 4 sensors-24-04353-f004:**
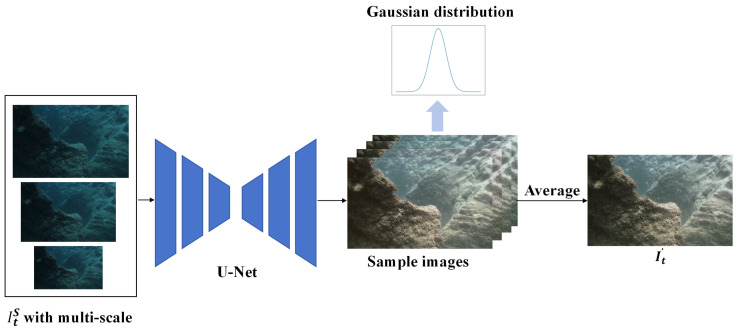
The specific workflow of the MC-IEM.

**Figure 5 sensors-24-04353-f005:**
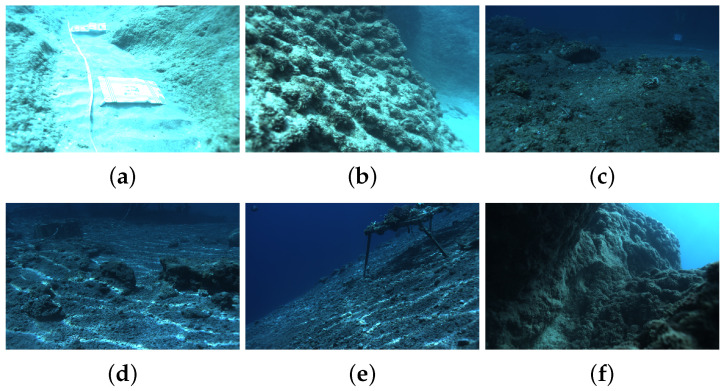
Examples from FLSea [47]. (**a**) Flatiron. (**b**) U canyon. (**c**) Northeast path. (**d**) Pier path. (**e**) Dice path. (**f**) Tiny canyon.

**Figure 6 sensors-24-04353-f006:**
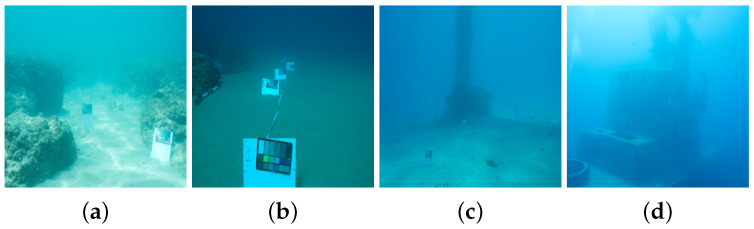
Examples from SQUID [32]. (**a**) Nachsholim. (**b**) Michmoret. (**c**) Katzaa. (**d**) Satil.

**Figure 7 sensors-24-04353-f007:**
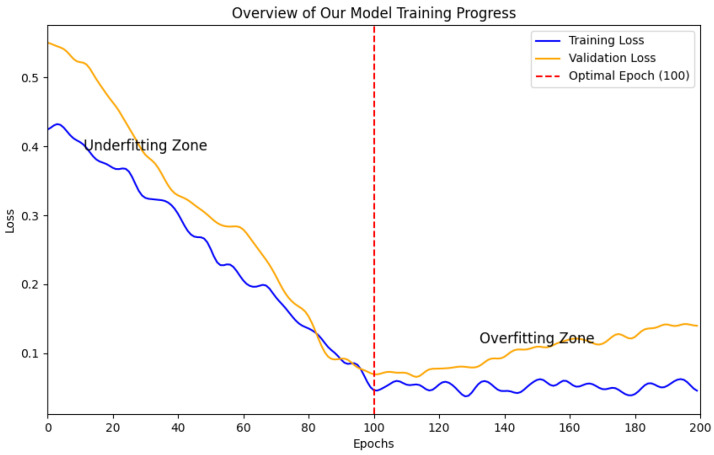
Overview of our model training progress.

**Figure 8 sensors-24-04353-f008:**
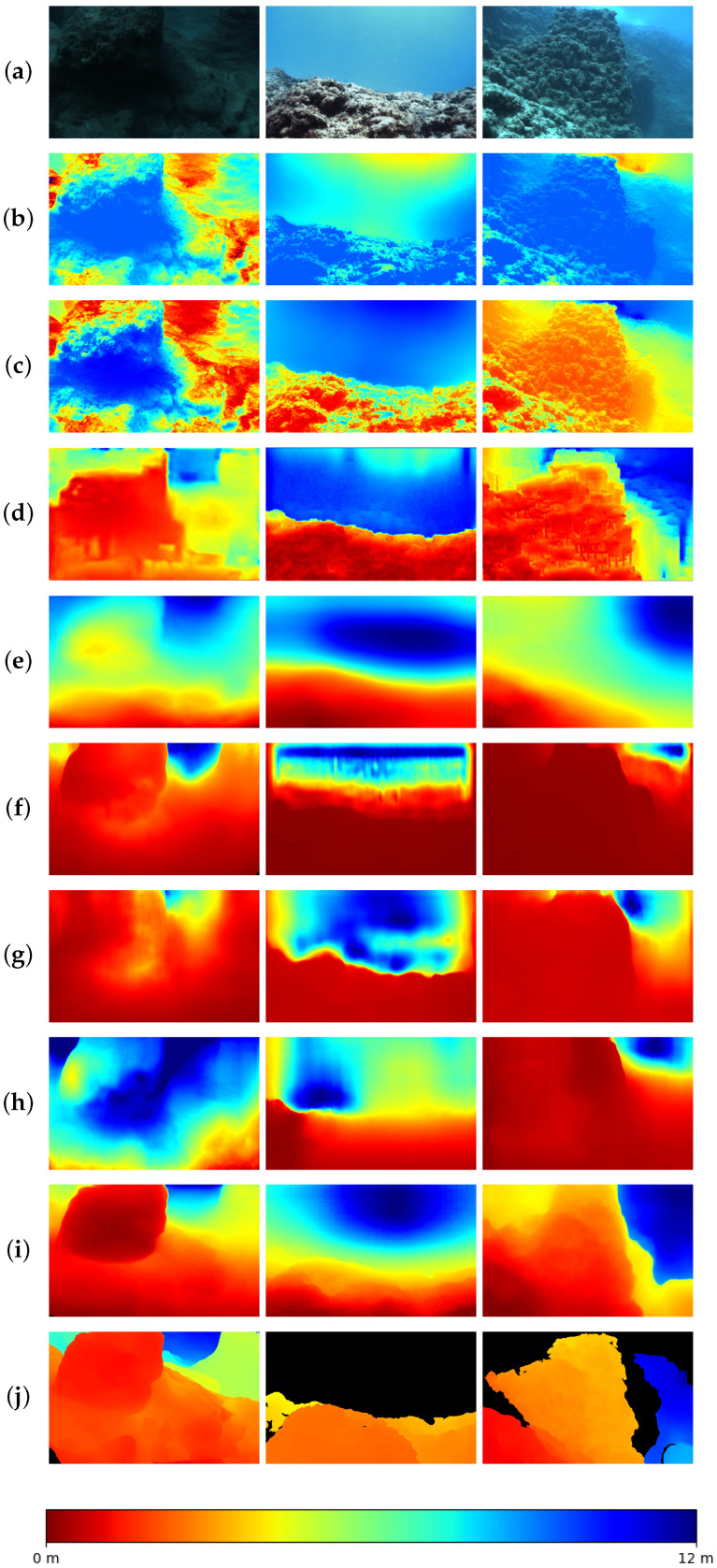
Qualitative comparison results on FLSea [47]. (**a**) Raw images. (**b**) DCP [12]. (**c**) UDCP [15]. (**d**) UW-Net [14]. (**e**) SC-Depth V3 [38]. (**f**) MonoViT [21]. (**g**) Lite-Mono [22]. (**h**) Robust-Depth [23]. (**i**) Ours. (**j**) Ground truth.

**Figure 9 sensors-24-04353-f009:**
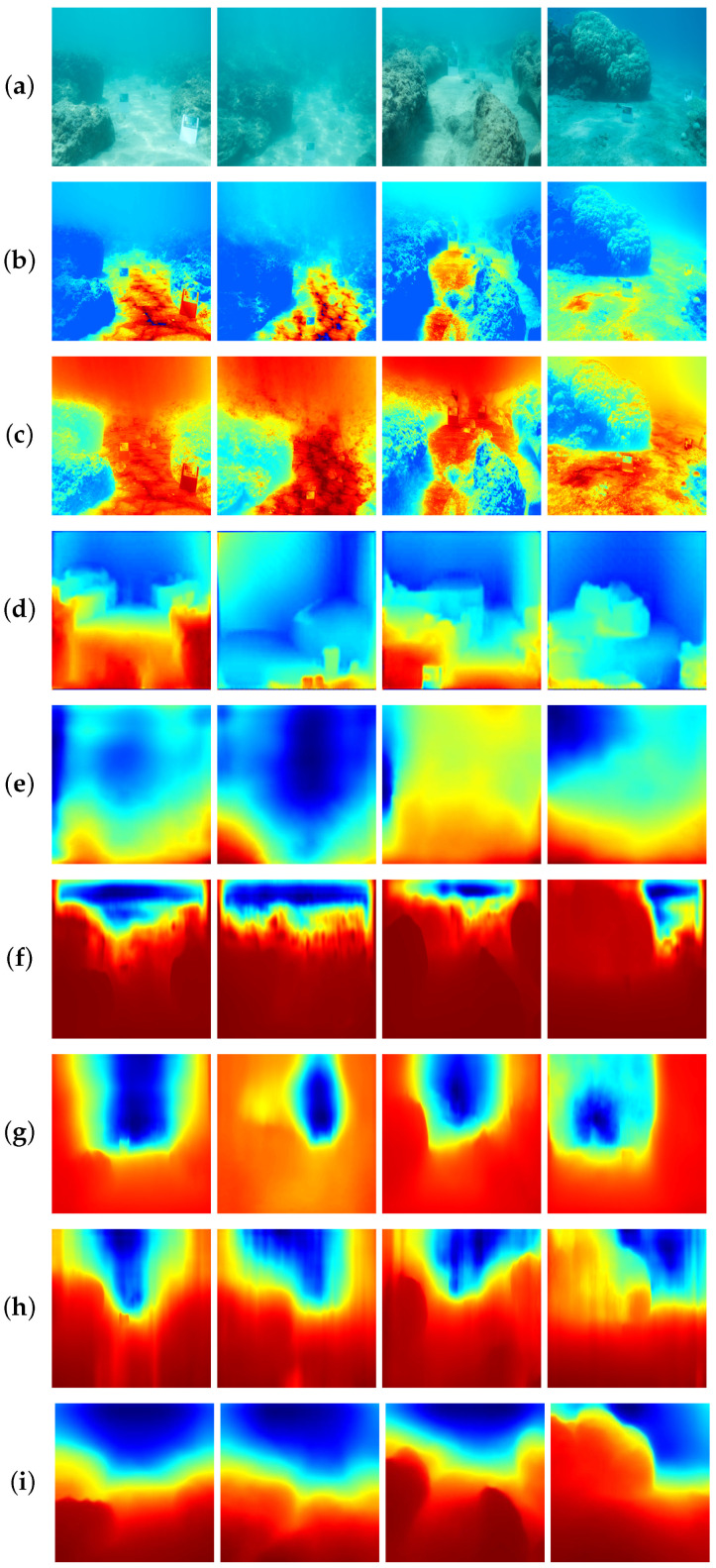
Qualitative comparison results on SQUID [32]. (**a**) Raw images. (**b**) DCP [12]. (**c**) UDCP [15]. (**d**) UW-Net [14]. (**e**) SC-Depth V3 [38]. (**f**) MonoViT [21]. (**g**) Lite-Mono [22]. (**h**) Robust-Depth [23]. (**i**) Ours. (**j**) Ground truth.

**Figure 10 sensors-24-04353-f010:**
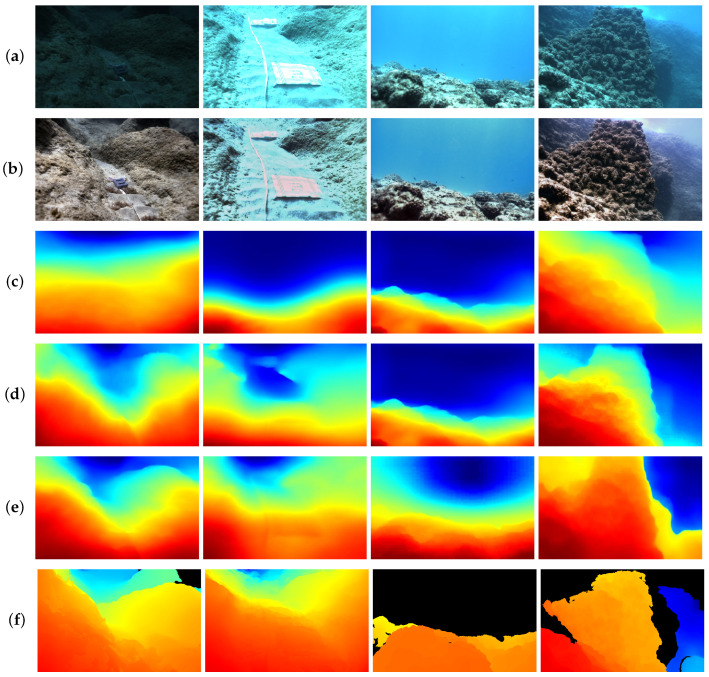
Qualitative comparison results of our ablation study on FLSea [47]. (**a**) Raw images. (**b**) Enhanced images. (**c**) Baseline. (**d**) Baseline+MC-IEM. (**e**) Ours. (**f**) Ground truth.

**Figure 11 sensors-24-04353-f011:**
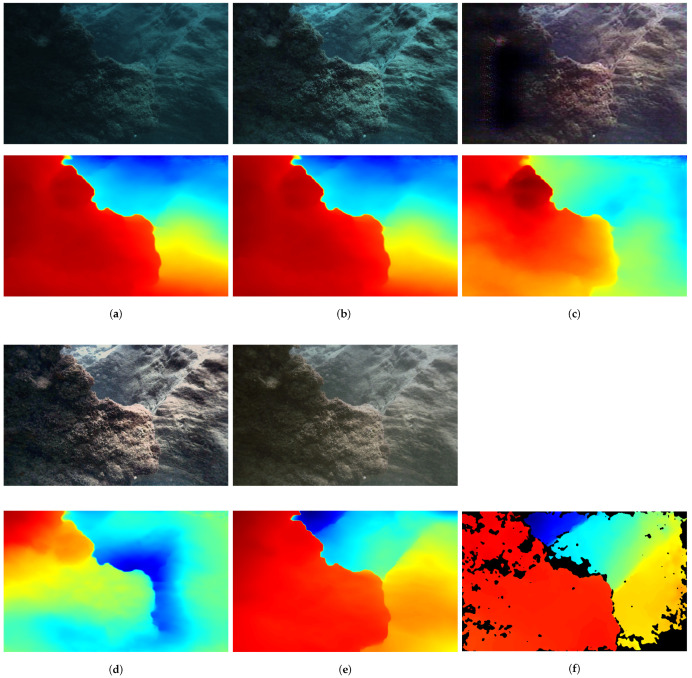
Qualitative comparison results on FLSea [47]. (**a**) Baseline + MIDAS. (**b**) Baseline + MIDAS + CLAHE [53]. (**c**) Baseline + MIDAS + FUnIE-GAN [54]. (**d**) Baseline + MIDAS + Water-Net [29]. (**e**) Ours. (**f**) Ground truth.

**Table 1 sensors-24-04353-t001:** Quantitative comparison results on FLSea [47] set.

Method	Error ↓				Accuracy ↑		
	AbsRel	Log10	RMSElog		δ<1.25	$\delta <1.25^{2}$	$\delta <1.25^{3}$
DCP [12]	1.527	0.402	1.243		0.207	0.356	0.489
UDCP [15]	0.577	0.217	0.646		0.337	0.575	0.731
UW-Net [14]	0.502	0.207	0.648		0.366	0.615	0.760
SC-Depth V3 [38]	0.500	0.233	0.730		0.306	0.550	0.728
MonoViT [21]	0.482	0.336	1.310		0.370	0.606	0.769
Lite-Mono [22]	0.379	0.136	**0.408**		0.502	0.774	**0.894**
Robust-Depth [23]	0.463	0.204	0.644		0.340	0.592	0.769
Ours	**0.239**	**0.132**	0.496		**0.588**	**0.819**	0.891

The best results in each category are in **bold**. ↓ signifies better performance with smaller errors, while ↑ signifies better performance with higher accuracy.

**Table 2 sensors-24-04353-t002:** Quantitative comparison results on SQUID [32].

Method	Error ↓				Accuracy ↑		
	AbsRel	Log10	RMSElog		δ<1.25	$\delta <1.25^{2}$	$\delta <1.25^{3}$
DCP [12]	3.641	0.410	1.240		0.177	0.343	0.479
UDCP [15]	1.827	0.371	1.090		0.183	0.346	0.487
UW-Net [14]	1.262	0.315	0.954		0.224	0.417	0.573
SC-Depth V3 [39]	1.044	0.297	0.901		0.234	0.440	0.596
MonoViT [21]	1.044	0.315	1.085		0.263	0.481	0.625
Lite-Mono [22]	1.426	0.328	0.981		0.211	0.404	0.550
Robust-Depth [23]	0.762	0.218	0.729		0.367	0.614	0.763
Ours	**0.476**	**0.172**	**0.623**		**0.469**	**0.731**	**0.845**

The best results in each category are in **bold**. ↓ signifies better performance with smaller errors, while ↑ signifies better performance with higher accuracy.

**Table 3 sensors-24-04353-t003:** Quantitative comparison results of our ablation study on FLSea [47].

Ablation Section			Evaluation Criteria						
MC-IEM	ADM		Error ↓				Accuracy ↑		
			AbsRel	Log10	RMSElog		δ<1.25	$\delta <1.25^{2}$	$\delta <1.25^{3}$
			0.367	0.177	0.572		0.402	0.675	0.826
✔			0.326	0.168	0.574		0.447	0.711	0.845
✔	✔		**0.239**	**0.132**	**0.496**		**0.588**	**0.819**	**0.891**

The best results in each category are in **bold**. ↓ signifies better performance with smaller errors, while ↑ signifies better performance with higher accuracy.

**Table 4 sensors-24-04353-t004:** Quantitative comparison results on FLSea [47].

Method	Error ↓		
	AbsRel	Log10	RMSElog
DPT [50]	0.851	0.875	2.084
LeRes [51]	0.786	0.739	1.180
MIDAS [25]	0.429	0.289	0.855
Ours	**0.239**	**0.132**	**0.496**

The best results in each category are in **bold**. ↓ signifies better performance with smaller errors, while ↑ signifies better performance with higher accuracy.

**Table 5 sensors-24-04353-t005:** Quantitative comparison results on FLSea [47].

Method	Error ↓				Accuracy ↑		
	AbsRel	Log10	RMSElog		δ<1.25	$\delta <1.25^{2}$	$\delta <1.25^{3}$
CLAHE [53]	0.752	0.660	1.581		0.014	0.028	0.044
FUnIE-GAN [54]	0.424	0.281	0.943		0.278	0.530	0.692
Water-Net [29]	0.431	0.321	1.042		0.248	0.457	0.591
Ours	**0.239**	**0.132**	**0.496**		**0.588**	**0.819**	**0.891**

The best results in each category are in **bold**. ↓ signifies better performance with smaller errors, while ↑ signifies better performance with higher accuracy.

## Data Availability

The experiments are evaluated on publicly open datasets. The datasets can be accessed in their corresponding published papers.

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
