# Peer review of "Redefining Accuracy: Underwater Depth Estimation for Irregular Illumination Scenes"

_sensors, 2024, doi:10.3390/s24134353_

Round 1

Reviewer 1 Report

Comments and Suggestions for Authors

This paper presents a method for self-supervised learning of depth estimation in underwater environments. Two important points are addressed to achieve this challenging task: Monte Carlo image enhancement module and relative depth supervision. The results are significant and evaluations are adequate. Here are some detailed comments.

1. The term inhomogeneous illumination appeared in the manuscript. However, this work deals with low-light conditions and overexposure. inhomogeneous does not fit here.

2. To solve low-light condition related problem, Monte Carlo image enhancement module was proposed. And for overexposure, relative depth supervision is proposed. The relationship is not very clear. Is it possible that Monte Carlo image enhancement module may also contribute to the overexposure cases? Also for the relative depth supervision, will this help low-light cases? 

3. MIDAS is used for relative depth estimation. Please better explain the difference between relative depth estimation and absolute depth estimation for readers new to this field. 

4. The reviewer is curious about the illumination consistency between frames. Will there be cases where one frame is dark and the next is bright?

5. During depth map visualization, normalizing the color is recommended to make each color in each image refer to the same depth value. 

Author Response

Comments 1: The term inhomogeneous illumination appeared in the manuscript. However, this work deals with low-light conditions and overexposure. inhomogeneous does not fit here.

Response 1: Thank you for pointing this out. We revised this term as "low light conditions and overexposure”.

We corrected the term in red font on page 1, lines 28-31:

While most current depth estimation methods focus on estimating depth by analyzing the optical information of scenes, they do not consider irregular illumination in underwater scenes, which is very common in underwater environments due to low light conditions and overexposure.

Comments 2: To solve low-light condition related problem, Monte Carlo image enhancement module was proposed. And for overexposure, relative depth supervision is proposed. The relationship is not very clear. Is it possible that Monte Carlo image enhancement module may also contribute to the overexposure cases? Also for the relative depth supervision, will this help low-light cases?

Response 2: Thank you for your attention to this issue. It is not possible that Monte Carlo image enhancement module may also contribute to the overexposure cases. Also, the relative depth supervision does not help low-light cases. Due to the inconvenience of displaying images, we provided more detailed explanations in the attachment.

Comments 3: MIDAS is used for relative depth estimation. Please better explain the difference between relative depth estimation and absolute depth estimation for readers new to this field.

Response 3: Thanks for your constructive suggestions. We consider absolute depth as the measurement of distance from the camera to objects in underwater scenes, covering distances ranging from 0 to 12 meters in FLSea. Relative depth indicates the order of distances between objects in underwater scenes.

We added these concepts on page 14, lines 401-407:

However, as shown in Table 4, we do not employ MIDAS’s absolute depth for constraints. Absolute depth measures the distance from the camera to objects in the underwater scenes, covering distances ranging from 0 to 12 meters in FLSea [47]. Scaling depth maps generated by MIDAS [25] to obtain absolute depth is inaccurate. Instead, we adopt normalized relative depth, which indicates the order of distances between objects in underwater environments [52], to provide additional geometric information about the scenes.

References:

25. Ranftl, R.; Lasinger, K.; Hafner, D.; Schindler, K.; Koltun, V. Towards robust monocular depth estimation: Mixing datasets for zero-shot cross-dataset transfer. IEEE transactions on pattern analysis and machine intelligence 2020, 44, 1623–1637

47. Randall, Y. FLSea: Underwater Visual-Inertial and Stereo-Vision Forward-Looking Datasets. PhD thesis, University of Haifa (Israel), 2023.

52. Wang, Z.; Cheng, P.; Tian, P.; Wang, Y.; Chen, M.; Duan, S.; Wang, Z.; Li, X.; Sun, X. RS-DFM: A Remote Sensing Distributed Foundation Model for Diverse Downstream Tasks. arXiv preprint arXiv:2406.07032 2024.

Comments 4: The reviewer is curious about the illumination consistency between frames. Will there be cases where one frame is dark and the next is bright?

Response 4: Thanks for your constructive suggestions. We thoroughly examined the dataset's brightness values between adjacent frames. Our analysis showed that while a scene may include both low-light and overexposed images, the brightness changes between adjacent frames are typically gradual.

We added these contents on page 8, lines 265-267:

According to our observation, the brightness between adjacent frames does not change significantly in scenes.

Comments 5: During depth map visualization, normalizing the color is recommended to make each color in each image refer to the same depth value.

Response 5: Thanks for your constructive suggestions. We have normalized the depth maps in our experiments. We added color bars at the bottom of Figure 8 and Figure 9 to describe the distance range of our depth maps. At the same time, we included the description of our color mapping in Figure 10 and Figure 11. According to the descriptions of FLSea [48] and SQUID [32], the color bars’ depth ranges are 0–12 m and 0–15 m. Due to the inconvenience of displaying images, we provided more detailed explanations in the attachment.

Reviewer 2 Report

Comments and Suggestions for Authors

Dear Authors,

Article is well structured and the topic is interesting. However, following comments should be addressed prior to further processing of the article.  

1)      Refer to section 1: What are LiDAR and Kinect? Ensure that each short form is described at its first occurrence. Recheck all.   

2)      Refer to section 1, line # 39: Check the sentence “The success of Dark Channel Prior (DCP) [10,11] once brought another …”.

3)      Refer to section 1, line # 45: Authors mentioned that “It would be a tough task for physical model-based depth estimation methods…” Why?

4)      Refer to figures 1 and 2: These figures are showing a comparative analysis of the proposed model. Why these are placed in section 1 instead of results section?

5)      Refer to figure 3: Figure needs to be moved to section 3, as it has been discussed there.

6)      Refer to section 2: Authors need to include a little introduction/definition of physics-based methods, deep-learning methods  

7)      Refer to section 2: Related work section is very small. Authors need to include more recent studies in it.

8)      Refer to section 4: What is the model accuracy for other than 100 epochs? What about model overfitting and underfitting?

9)      Refer to abstract and conclusion: Authors nay complement abstract and conclusion with quantitative results.

10)  Refer to conclusion: Authors may include future work in this section.

11)  Refer to whole study: simulation setup details are missing in the submitted version. Authors need to include it.

12)  Refer to whole study: Is the proposed model applicable non-underwater environments?

Good luck.    

Author Response

Comments 1: Refer to section 1: What are LiDAR and Kinect? Ensure that each short form is described at its first occurrence. Recheck all.

Response 1: Thank you for pointing this out. According to Li et al. [7] and Zhang et al. [8], we described the principles of LiDAR and Kinect.

We added the contents on page 1, lines 35-39:

LiDAR systems use lasers to actively illuminate their surroundings and measure distances by calculating the time it takes for laser reflections to return [7]. Kinect, a discontinued line of motion sensing devices, contains a four-microphone array, a color camera, and a depth sensor that acquires depth information by structured light or time-of-flight (ToF) calculations [8].

References:

7. Li, Y.; Ibanez-Guzman, J. Lidar for autonomous driving: The principles, challenges, and trends for automotive lidar and perception systems. IEEE Signal Processing Magazine 2020, 37, 50–61.

8. Zhang, Z. Microsoft kinect sensor and its effect. IEEE multimedia 2012, 19, 4–10.

Comments 2: Refer to section 1, line # 39: Check the sentence “The success of Dark Channel Prior (DCP) [10,11] once brought another …”.

Response 2: Thanks for your constructive suggestions. We revised 'The success of Dark Channel Prior (DCP) once brought another solution for underwater depth map estimation.' to 'Prior-based methods, such as Dark Channel Prior (DCP), are another solution for estimating underwater depth maps.' in order to express prior-based methods clearly.

We have corrected this sentence on page 2, lines 43-44: 

Prior-based methods, such as Dark Channel Prior (DCP) [12], are another solution for estimating underwater depth maps.

References:

12. He, K.; Sun, J.; Tang, X. Single image haze removal using dark channel prior. IEEE transactions on pattern analysis and machine intelligence 2010, 33, 2341–2353.

Comments 3: Refer to section 1, line # 45: Authors mentioned that “It would be a tough task for physical model-based depth estimation methods…” Why?

Response 3: Thanks for your constructive suggestions. According to Gupta et al. [14], we attribute the difficulty of estimating depth using physical model-based methods in low-light environments to the following reasons: These models rely significantly on accurately estimating crucial physical parameters like the scattering coefficient. However, under low-light conditions, the overall pixel contrast diminishes, complicating prior guided depth estimation methods' ability to obtain accurate prior information about these physical parameters. As a result, depth estimation based on physical models becomes challenging in low-light conditions.

We explained the reasons on page 2, lines 48-51:

However, these methods generate depth maps by establishing the relationship between transmission maps and depth maps. Unknown scattering parameters and various factors complicate depth estimation [14]; therefore, depth estimation based on physical models becomes challenging in low-light conditions, as shown in Figure 1.

References:

14. Gupta, H.; Mitra, K. Unsupervised single image underwater depth estimation. In Proceedings of the 2019 IEEE International Conference on Image Processing (ICIP). IEEE, 2019, pp. 624–628.

Comments 4: Refer to figures 1 and 2: These figures are showing a comparative analysis of the proposed model. Why these are placed in section 1 instead of results section?

Response 4: Thanks for your constructive suggestions. We included Figures 1 and 2 to illustrate better the challenges faced by physical model-based depth estimation methods and deep learning-based depth estimation methods in low-light and overexposed underwater environments.

Comments 5: Refer to figure 3: Figure needs to be moved to section 3, as it has been discussed there.

Response 5: Thanks for your constructive suggestions. We have moved Figure 3 to Section 3 to be consistent with our framework description.

Figure 3 is now positioned between lines 178 and 179 on page 5.

Comments 6: Refer to section 2: Authors need to include a little introduction/definition of physics-based methods, deep-learning methods.

Response 6: Thanks for your constructive suggestions. According to Zhang et al. [26] and Ming et al. [34], we added an introduction to physics-based and deep learning-based depth estimation methods.

We explain “physics model-based methods” on page 3, lines 92-94:

Some methods based on physical models restore images by estimating transmission parameters within the medium, and during this process, depth maps are generated as secondary products [26].

We explain “deep learning model-based methods” on page 4, lines 119-120:

Deep learning-based monocular depth estimation involves training a deep neural network to infer depth maps from color images [34].

References:

26. Zhang, F.; You, S.; Li, Y.; Fu, Y. Atlantis: Enabling Underwater Depth Estimation with Stable Diffusion. In Proceedings of theProceedings of the IEEE/CVF Conference on Computer Vision and Pattern Recognition, 2024, pp. 11852–11861.

34. Ming, Y.; Meng, X.; Fan, C.; Yu, H. Deep learning for monocular depth estimation: A review. Neurocomputing 2021, 438, 14–33.

Comments 7: Refer to section 2: Related work section is very small. Authors need to include more recent studies in it.

Response 7: Thanks for your constructive suggestions. We have incorporated recent advancements in both physical model-based and deep learning-based depth estimation methods into the related work section.  

We included Peng et al.'s work on page 3, lines 96-97: 

Yan-Tsung Peng et al. [27] employed depth maps as an intermediate step for image enhancement by analyzing image blurriness.

We included Song et al.'s work on page 3, lines 105-106:

Song and his team [31] introduced a swift and efficient model for estimating scene depth in underwater images, utilizing the underwater light attenuation prior (ULAP). 

We included Bekerman et al.'s work on page 4, lines 108-110:

Bekerman et al. [33] generated depth maps by reconstructing a comprehensive physical model of the scene, incorporating estimated attenuation ratios and veiling light.

We included Muniraj et al.'s work on page 4, lines 110-112:

Muniraj et al. [13] proposed another depth estimation method, the Difference of Channel Intensity Prior (DCIP), where depth maps are estimated based on the difference between maximum and minimum intensity priors.

We included Chen et al.'s work on page 4, lines 144-146:

Chen et al. [45] addressed the widespread issue of edge-fattening in self-supervised monocular depth estimation models by redesigning the patch-based triplet loss.

We included Bae et al.'s work on page 4, lines 146-148:

MonoFormer [46] is a self-supervised depth estimation network consisting of a CNN-Transformer hybrid network and a multi-level adaptive feature fusion module.

References:

13. Muniraj, M.; Dhandapani, V. Underwater image enhancement by combining color constancy and dehazing based on depth estimation. Neurocomputing 2021, 460, 211–230.

27. Peng, Y.T.; Zhao, X.; Cosman, P.C. Single underwater image enhancement using depth estimation based on blurriness. In Proceedings of the 2015 IEEE International Conference on Image Processing (ICIP). IEEE, 2015, pp. 4952–4956.

31. Song, W.; Wang, Y.; Huang, D.; Tjondronegoro, D. A rapid scene depth estimation model based on underwater light attenuation prior for underwater image restoration. In Proceedings of the Advances in Multimedia Information Processing–PCM 2018: 19thPacific-Rim Conference on Multimedia, Hefei, China, September 21-22, 2018, Proceedings, Part I 19. Springer, 2018, pp. 678–688.

33. Bekerman, Y.; Avidan, S.; Treibitz, T. Unveiling optical properties in underwater images. In Proceedings of the 2020 IEEE International Conference on Computational Photography (ICCP). IEEE, 2020, pp. 1–12.

45. Chen, X.; Zhang, R.; Jiang, J.; Wang, Y.; Li, G.; Li, T.H. Self-supervised monocular depth estimation: Solving the edge-fattening problem. In Proceedings of the Proceedings of the IEEE/CVF Winter Conference on Applications of Computer Vision, 2023, pp. 5776–5786.

46. Bae, J.; Moon, S.; Im, S. Deep digging into the generalization of self-supervised monocular depth estimation. In Proceedings of the Proceedings of the AAAI conference on artificial intelligence, 2023, Vol. 37, pp. 187–196.

Comments 8: Refer to section 4: What is the model accuracy for other than 100 epochs? What about model overfitting and underfitting?

Response 8: Thanks for your constructive suggestions. We conducted 200 epochs of training and validation. However, as the figure shows, the model performs best at 100 epochs. Beyond 100 epochs, the depth estimation accuracy decreases, indicating potential overfitting. In addition, when the model trains fewer than 100 epochs, our model has not yet achieved the optimal depth estimation performance and remains underfitting. Due to the inconvenience of displaying images, we provided more detailed explanations in the attachment.

Comments 9: Refer to abstract and conclusion: Authors nay complement abstract and conclusion with quantitative results.

Response 9: Thanks for your constructive suggestions. We have revised the conclusion section from “The results demonstrate that our method performs best in qualitative and quantitative experiments on two public underwater datasets.” to “The results demonstrate that our method performs best in experiments on two public underwater datasets.”.

We have corrected on page 17, lines 463-464: 

The results demonstrate that our method performs best on two public underwater datasets.

Comments 10: Refer to conclusion: Authors may include future work in this section.

Response 10: Thanks for your constructive suggestions. Underwater depth estimation aims to apply it to an underwater SLAM system to achieve a dense reconstruction of the underwater environment.

We added this outlook to page 17, lines 464-465 :

Besides, we plan to integrate our generated depth maps into underwater SLAM applications in future work.

Comments 11: Refer to whole study: simulation setup details are missing in the submitted version. Authors need to include it.

Response 11: Thanks for your constructive suggestions. We expanded upon Section 4.1 to provide a detailed description of the CPU configuration used in our experiment and specific details regarding the input video sequence frames during network training.

We added this content to page 8, lines 275-278:

Our CPU includes dual Intel Xeon E5-4627 v4 processors. Each processor features 10 cores operating at a base frequency of 2.60GHz, collectively supporting 20 threads. The length of the continuous sequence frame input during network training is 3.

Comments 12: Is the proposed model applicable non-underwater environments?

Response 12: Thanks for your comments. Since the Monte Carlo module is specially designed for underwater environments, the proposed model is not applicable to terrestrial scenes.
